# Enhanced LSTM for Natural Language Inference

## Abstract

Reasoning and inference are central to human and artificial intelligence. Modeling inference in human language is very challenging. With the availability of large annotated data (Bowman et al., 2015), it has recently become feasible to train neural network based inference models, which have shown to be very effective. In this paper, we present a new state-of-the-art result, achieving the accuracy of 88.6% on the Stanford natural language inference dataset. Unlike the previous top models that use very complicated network architectures, we first demonstrate that carefully designing sequential inference models based on chain LSTMs can outperform all previous models. Based on this, we further show that by explicitly considering recursive architectures in both local inference modeling and inference composition, we achieve additional improvement. Particularly, incorporating syntactic parsing information contributes to our best result—it further improves the performance even when added to the already very strong model.

## 1 Introduction

Reasoning and inference are central to both human and artificial intelligence. Modeling inference in human language is notoriously challenging but is a basic problem towards true natural language understanding, as pointed out by MacCartney and Manning (2008), *"a necessary (if not sufficient) condition for true natural language understanding is a mastery of open-domain natural language inference."* The efforts have also included a large bulk of work on recognizing textual entailment.

Specifically, natural language inference (NLI) is concerned with determining whether a natural-language hypothesis $h$ can be inferred from a premise $p$, as depicted in the following example from MacCartney (2009), where the hypothesis is regarded to be entailed from the premise.

p: *Several airlines polled saw costs grow more than expected, even after adjusting for inflation.*

h: *Some of the companies in the poll reported cost increases.*

The most recent years have seen advances in modeling natural language inference. An important contribution is the creation of a much larger annotated dataset, the Stanford Natural Language Inference (SNLI) dataset (Bowman et al., 2015). The corpus has 570,000 human-written English sentence pairs manually labeled by multiple human subjects. This makes it feasible to train more complex inference models. Neural network models, which often need relatively large annotated data to estimate their parameters, have shown to achieve the state of the art on SNLI (Bowman et al., 2015, 2016; Munkhdalai and Yu, 2016b; Parikh et al., 2016; Sha et al., 2016; Paria et al., 2016).

While some previous top-performing models use rather complicated network architectures to achieve the state-of-the-art results (Munkhdalai and Yu, 2016b), we demonstrate in this paper that enhancing sequential inference models based on chain models can outperform all previous results, suggesting that the potential of such sequential inference approaches have not been fully exploited yet. Our model may serve as a new baseline or starting point for deploying more complicated models for NLI. More specifically, we show that our sequential inference model achieves an accuracy of 88.0% on the SNLI benchmark.

Exploring syntax for NLI is very attractive to us. In many problems, syntax and semantics interact closely, as generally phrased in the slogan "the syntax and the semantics work together in tandem" (Barker and Jacobson, 2007), among others. Complicated tasks such as natural language

inference could well involve both, which has been discussed in the context of recognizing textual entailment (RTE) (Mehdad et al., 2010; Ferrone and Zanzotto, 2014). In this paper, we are interested in exploring this within the neural network frameworks, with the presence of relatively large training data. We show that by explicitly encoding parsing information with recursive networks in both local inference modeling and inference composition, and incorporating it into our framework, we achieve additional improvement, increasing the performance to a new state of the art with a 88.6% accuracy.

## 2   Related Work

Early work on natural language inference has been performed on rather small datasets with more conventional methods (refer to MacCartney (2009) for a good literature survey), which includes a large bulk of early work on recognizing textual entailment, such as (Dagan et al., 2005; Iftene and Balahur-Dobrescu, 2007), among others. More recently, Bowman et al. (2015) made available the SNLI dataset with the 570,000 human annotated sentence pairs. They also experimented with simple classification models as well as simple neural networks that encode the premise and hypothesis independently. Rocktäschel et al. (2015) proposed neural attention-based models for NLI, which captured the attention information. In general, attention based models have been shown to be effective in a wide range of tasks, including machine translation (Bahdanau et al., 2014), speech recognition (Chorowski et al., 2015; Chan et al., 2016), image caption (Xu et al., 2015), and text summarization (Rush et al., 2015; Chen et al., 2016), among others. For NLI, the idea allows neural models to pay attention to specific areas of the sentences.

A variety of more advanced networks have been developed since then (Bowman et al., 2016; Vendrov et al., 2015; Mou et al., 2016; Liu et al., 2016; Munkhdalai and Yu, 2016a), and inter-sentence attention-based models (Rocktäschel et al., 2015; Wang and Jiang, 2016; Cheng et al., 2016; Parikh et al., 2016; Munkhdalai and Yu, 2016b; Sha et al., 2016; Paria et al., 2016). Sha et al. (2016) proposes a special LSTM variant which considers the attention vector of another sentence as an inner state of LSTM. Paria et al. (2016) use a neural architecture with a complete binary tree-LSTM encoders while without syntactic information. Among them, more relevant to our work are the approaches proposed by Parikh et al. (2016) and Munkhdalai and Yu (2016b), which are among the best models.

Parikh et al. (2016) propose a relatively simple but very effective decomposable model. The model decomposes the NLI problem into subproblems that can be solved separately. On the other hand, Munkhdalai and Yu (2016b) propose much more complicated networks that consider sequential LSTM-based encoding, recursive networks, and complicated combinations of attention models, which provide about 0.5% gain over the results reported by Parikh et al. (2016).

It is, however, not very clear if the potential of the sequential inference networks has been well exploited for NLI. In this paper, we first revisit this problem and show that enhancing sequential inference models based on chain networks can actually outperform all previous results. Our model may serve as a new baseline or starting point for deploying more complicated models for NLI. We further show that explicitly considering recursive architectures to encode syntactic parsing information for NLI could further improve the performance.

## 3   Hybrid Neural Inference Models

We present here our natural language inference networks which are composed of the following major components: input encoding, local inference modeling, and inference composition. Figure 1 shows a high-level view of the architecture. Vertically, the figure depicts the three major components, and horizontally, the left side of the figure represents our sequential NLI model named ESIM, and the right side represents networks that incorporate syntactic parsing information in tree LSTMs.

In our notation, we have two sentences $\mathbf{a} = (\mathbf{a}_1, \ldots, \mathbf{a}_{\ell_a})$ and $\mathbf{b} = (\mathbf{b}_1, \ldots, \mathbf{b}_{\ell_b})$, where $\mathbf{a}$ is a premise and $\mathbf{b}$ a hypothesis. The $\mathbf{a}_i$ or $\mathbf{b}_j \in \mathbb{R}^l$ is an embedding of $l$-dimensional vector, which can be initialized with some pre-trained word embeddings and organized with parse trees. The goal is to predict a label $y$ that indicates the logic relationship between $\mathbf{a}$ and $\mathbf{b}$.

### 3.1   Input Encoding

We employ bidirectional LSTM (BiLSTM) as one of our basic building blocks for NLI. We first use it to encode the input premise and hypothesis (Equation (1) and (2)). Here BiLSTM learns to represent a word (e.g., $\mathbf{a}_i$) and its context. Later we will also use BiLSTM to perform *inference composition* to

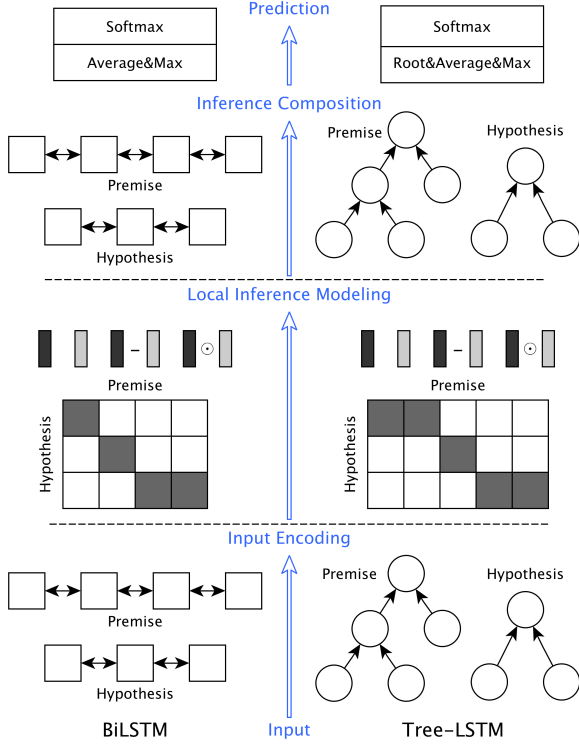

Figure 1: A high-level view of our hybrid neural inference networks.

construct the final prediction, where BiLSTM encodes local inference information and its interaction. To bookkeep the notations for later use, we write as $\bar{\mathbf{a}}_i$ the hidden (output) state generated by the BiLSTM at time $i$ over the input sequence $\mathbf{a}$. The same is applied to $\bar{\mathbf{b}}_j$:

$$\bar{\mathbf{a}}_i = \text{BiLSTM}(\mathbf{a}, i), \forall i \in [1, \dots, \ell_a], \qquad (1)$$

$$\bar{\mathbf{b}}_j = \text{BiLSTM}(\mathbf{b}, j), \forall j \in [1, \dots, \ell_b]. \qquad (2)$$

Due to the space limit, we will skip the description of the basic chain LSTM and readers can refer to Hochreiter and Schmidhuber (1997) for details. Briefly, when modeling a sequence, an LSTM employs a set of soft gates together with a memory cell to control message flows, resulting in an effective modeling of tracking long-distance information/dependencies in a sequence.

A bidirectional LSTM runs a forward and backward LSTM on a sequence starting from the left and the right end, respectively. The hidden states generated by these two LSTMs at each time step are concatenated to represent that time step and its context. Note that we used LSTM memory blocks in our models. We examined other recurrent memory blocks such as GRUs (Gated Recurrent Units) (Cho et al., 2014) and they are inferior to LSTMs on the heldout set for our NLI task.

As discussed above, it is intriguing to explore the effectiveness of syntax for natural language inference; for example, whether it is useful even when incorporated into the best-performing models. To this end, we will also encode syntactic parse trees of a premise and hypothesis through tree-LSTM (Zhu et al., 2015; Tai et al., 2015; Le and Zuidema, 2015), which extends the chain LSTM to a recursive network (Socher et al., 2011).

Specifically, given the parse of a premise or hypothesis, a tree node is deployed with a tree-LSTM memory block depicted as in Figure 2 and computed with Equations (3–10). In short, at each node, an input vector $\mathbf{x}_t$ and the hidden vectors of its two children (the left child $\mathbf{h}_{t-1}^L$ and the right child $\mathbf{h}_{t-1}^R$) are taken in as the input to calculate the current node's hidden vector $\mathbf{h}_t$.

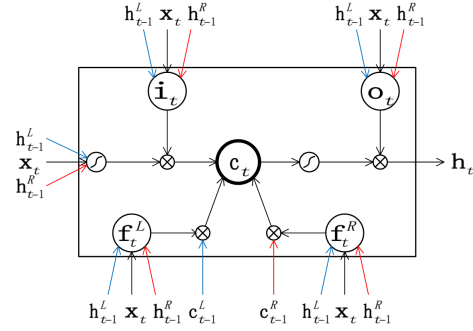

Figure 2: A tree-LSTM memory block.

We describe the updating of a node at a high level with Equation (3) to facilitate references later in the paper, and the detailed computation is described in (4–10). Specifically, the input of a node is used to configure four gates: the input gate $\mathbf{i}_t$, output gate $\mathbf{o}_t$, and the two forget gates $\mathbf{f}_t^L$ and $\mathbf{f}_t^R$. The memory cell $\mathbf{c}_t$ considers each child's cell vector, $\mathbf{c}_{t-1}^L$ and $\mathbf{c}_{t-1}^R$, which are gated by the left forget gate $\mathbf{f}_t^L$ and right forget gate $\mathbf{f}_t^R$, respectively.

$$\mathbf{h}_t = \text{TrLSTM}(\mathbf{x}_t, \mathbf{h}_{t-1}^L, \mathbf{h}_{t-1}^R), \qquad (3)$$

$$\mathbf{h}_t = \mathbf{o}_t \odot \tanh(\mathbf{c}_t), \qquad (4)$$

$$\mathbf{o}_t = \sigma(\mathbf{W}_o\mathbf{x}_t + \mathbf{U}_o^L\mathbf{h}_{t-1}^L + \mathbf{U}_o^R\mathbf{h}_{t-1}^R), \qquad (5)$$

$$\mathbf{c}_t = \mathbf{f}_t^L \odot \mathbf{c}_{t-1}^L + \mathbf{f}_t^R \odot \mathbf{c}_{t-1}^R + \mathbf{i}_t \odot \mathbf{u}_t, \qquad (6)$$

$$\mathbf{f}_t^L = \sigma(\mathbf{W}_f\mathbf{x}_t + \mathbf{U}_f^{LL}\mathbf{h}_{t-1}^L + \mathbf{U}_f^{LR}\mathbf{h}_{t-1}^R), \qquad (7)$$

$$\mathbf{f}_t^R = \sigma(\mathbf{W}_f\mathbf{x}_t + \mathbf{U}_f^{RL}\mathbf{h}_{t-1}^L + \mathbf{U}_f^{RR}\mathbf{h}_{t-1}^R), \qquad (8)$$

$$\mathbf{i}_t = \sigma(\mathbf{W}_i\mathbf{x}_t + \mathbf{U}_i^L\mathbf{h}_{t-1}^L + \mathbf{U}_i^R\mathbf{h}_{t-1}^R), \qquad (9)$$

$$\mathbf{u}_t = \tanh(\mathbf{W}_c\mathbf{x}_t + \mathbf{U}_c^L\mathbf{h}_{t-1}^L + \mathbf{U}_c^R\mathbf{h}_{t-1}^R), \qquad (10)$$

where $\sigma$ is the sigmoid function, $\odot$ is the element-wise multiplication of two vectors, and all $\mathbf{W} \in \mathbb{R}^{d \times l}$, $\mathbf{U} \in \mathbb{R}^{d \times d}$ are weight matrices to be learned.

In the current *input encoding* layer, $\mathbf{x}_t$ is used to encode a word embedding for a leaf node. Since a non-leaf node does not correspond to a specific word, we use a special vector $\mathbf{x}_t'$ as its input, which is like an unknown word. However, in the *inference composition* layer that we discuss later, the goal of using tree-LSTM is very different and the input $\mathbf{x}_t$ will be very different as well—it will encode local inference information and will have values at all tree nodes.

### 3.2 Local Inference Modeling

Carefully modeling local subsentential inference between a premise and hypothesis is critical to help determine the overall inference between these two statements.

To closely examine local inference, we explore both the sequential and syntactic tree models that have been discussed above. The former helps collect local inference for words and their context, and the tree LSTM helps collect local information between (linguistic) phrases and clauses.

**Locality of inference** Modeling local inference needs to employ some forms of hard or soft alignment to associate the relevant subcomponents between a premise and a hypothesis. This includes early methods motivated from the alignment in conventional automatic machine translation (MacCartney, 2009). In neural network models, this is often achieved with soft attention.

Parikh et al. (2016) decomposed this process: the word sequence of the premise (or hypothesis) is regarded as a bag-of-word embedding vector and inter-sentence "alignment" (or attention) is computed individually to softly align each word to the content of hypothesis (or premise, respectively). While their basic framework is very effective, achieving one of the previous best results, using a pre-trained word embedding by itself does not automatically consider the context around a word in NLI. Parikh et al. (2016) did take into account the word order and context information through an optional distance-sensitive intra-sentence attention.

In this paper, we argue for leveraging attention over the bidirectional sequential encoding of the input, as discussed above. We will show that this plays an important role in achieving our best results, and the intra-sentence attention used by (Parikh et al., 2016) actually does not further improve over our model, while the overall framework they proposed is very effective.

Our soft alignment layer computes the attention weights as the similarity of a hidden state tuple $<\bar{\mathbf{a}}_i, \bar{\mathbf{b}}_j>$ between a premise and a hypothesis with Equation (11). We did study more complicated relationships between $\bar{\mathbf{a}}_i$ and $\bar{\mathbf{b}}_j$ with multilayer perceptrons, but observed no further improvement on the heldout data.

$$e_{ij} = \bar{\mathbf{a}}_i^T \bar{\mathbf{b}}_j \qquad (11)$$

$\bar{\mathbf{a}}_i$ and $\bar{\mathbf{b}}_j$ are computed above in Equations (1) and (2), or with Equation (3) when tree-LSTM is used. Again, as discussed above, we will use bidirectional LSTM and tree-LSTM to encode the premise and hypothesis, respectively. In our sequential inference model, unlike in Parikh et al. (2016) which proposed to use a function $F(\bar{\mathbf{a}}_i)$, i.e., a feed-forward neural network, to map the original word representation for calculating $e_{ij}$, we instead advocate to use BiLSTM, which encodes the information in premise and hypothesis very well and achieves better performance shown in the experiment section; we tried to apply the $F(.)$ function on our hidden states before computing $e_{ij}$ and it did not further help our models.

**Local inference collected over sequences** Local inference is determined by the attention weight $e_{ij}$ computed above, which is used to obtain the local relevance between a premise and hypothesis. For the hidden state of a word in a premise, i.e., $\bar{\mathbf{a}}_i$ (already encoding the word itself and its context), the relevant semantics in the hypothesis is identified and composed using $e_{ij}$, more specifically with Equation (12).

$$\tilde{\mathbf{a}}_i = \sum_{j=1}^{\ell_b} \frac{\exp(e_{ij})}{\sum_{k=1}^{\ell_b} \exp(e_{ik})} \bar{\mathbf{b}}_j, \forall i \in [1, \dots, \ell_a], \quad (12)$$

$$\tilde{\mathbf{b}}_j = \sum_{i=1}^{\ell_a} \frac{\exp(e_{ij})}{\sum_{k=1}^{\ell_a} \exp(e_{kj})} \bar{\mathbf{a}}_i, \forall j \in [1, \dots, \ell_b], \quad (13)$$

where $\tilde{\mathbf{a}}_i$ is a weighted summation of $\{\bar{\mathbf{b}}_j\}_{j=1}^{\ell_b}$. Intuitively, the content in $\{\bar{\mathbf{b}}_j\}_{j=1}^{\ell_b}$ that is relevant to $\bar{\mathbf{a}}_i$ will be selected and represented as $\tilde{\mathbf{a}}_i$. The same is performed for each word in the hypothesis with Equation (13).

**Local inference collected over parse trees** We use tree models to help collect local inference information over linguistic phrases and clauses in this layer. The tree structures of the premise and hypothesis are produced by a constituency parser.

Once the hidden states of a tree are all computed with Equation (3), we treat all tree nodes equally

as we do not have further heuristics to discriminate them, but leave the attention weights to figure out their relationship. So, we use Equation (11) to compute the attention weights for all node pairs between a premise and hypothesis. This connects all words, constituent phrases, and clauses between the premise and hypothesis. We then collect the information between all the pairs with Equations (12) and (13) and feed them into the next layer.

**Enhancement of local inference information** In our models, we further enhance the local inference information collected. We compute the difference and the element-wise product for the tuple $<\bar{\mathbf{a}}, \tilde{\mathbf{a}}>$ as well as for $<\bar{\mathbf{b}}, \tilde{\mathbf{b}}>$. We expect that such operations could help sharpen local inference information between elements in the tuples and capture inference relationships such as contradiction. The difference and element-wise product are then concatenated with the original vectors, $\bar{\mathbf{a}}$ and $\tilde{\mathbf{a}}$, or $\bar{\mathbf{b}}$ and $\tilde{\mathbf{b}}$, respectively. The enhancement is performed for both the sequential and the tree models.

$$\mathbf{m}_a = [\bar{\mathbf{a}}; \tilde{\mathbf{a}}; \bar{\mathbf{a}} - \tilde{\mathbf{a}}; \bar{\mathbf{a}} \odot \tilde{\mathbf{a}}], \quad (14)$$

$$\mathbf{m}_b = [\bar{\mathbf{b}}; \tilde{\mathbf{b}}; \bar{\mathbf{b}} - \tilde{\mathbf{b}}; \bar{\mathbf{b}} \odot \tilde{\mathbf{b}}]. \quad (15)$$

This process could be regarded as a special case of modeling some high-order interaction between the tuple elements. Along this direction, we have also further modeled the interaction by feeding the tuples into feed-forward networks and added the top layer hidden states to the above concatenation. We found that it does not further help the inference accuracy on the heldout dataset.

### 3.3 Inference Composition

To determine the overall inference relationship between a premise and hypothesis, we explore a composition layer to compose the enhanced local inference information $\mathbf{m}_a$ and $\mathbf{m}_b$. We perform the composition sequentially or in its parse context using BiLSTM and tree-LSTM, respectively.

**The composition layer** In our sequential inference model, we keep using BiLSTM to compose local inference information sequentially. The formulas for BiLSTM are similar to those in Equations (1) and (2) in their forms so we skip the details, but the aim is very different here—they are used to capture local inference information $\mathbf{m}_a$ and $\mathbf{m}_b$ and their context here for inference composition.

In the tree composition, the high-level formulas of how a tree node is updated to compose local inference is as follows:

$$\mathbf{v}_{a,t} = \text{TrLSTM}(F(\mathbf{m}_{a,t}), \mathbf{h}_{t-1}^L, \mathbf{h}_{t-1}^R), \quad (16)$$

$$\mathbf{v}_{b,t} = \text{TrLSTM}(F(\mathbf{m}_{b,t}), \mathbf{h}_{t-1}^L, \mathbf{h}_{t-1}^R). \quad (17)$$

We propose to control model complexity in this layer, since the concatenation we described above to compute $\mathbf{m}_a$ and $\mathbf{m}_b$ can significantly increase the overall parameter size to potentially overfit the models. We propose to use a mapping $F$ as in Equation (16) and (17). More specifically, we use a 1-layer feed-forward neural network with the ReLU activation. This function is also applied to BiLSTM in our sequential inference composition.

**Pooling** Our inference model converts the resulting vectors obtained above to a fixed-length vector with pooling and feeds it to the final classifier to determine the overall inference relationship.

We consider that summation (Parikh et al., 2016) could be sensitive to the sequence length and hence less robust. We instead suggest the following strategy: compute both average and max pooling, and concatenate all these vectors to form the final fixed length vector $\mathbf{v}$. Our experiments show that this leads to significantly better results than summation. The final fixed length vector $\mathbf{v}$ is calculated as follows:

$$\mathbf{v}_{a,\text{ave}} = \sum_{i=1}^{\ell_a} \frac{\mathbf{v}_{a,i}}{\ell_a}, \quad \mathbf{v}_{a,\max} = \max_{i=1}^{\ell_a} \mathbf{v}_{a,i}, \quad (18)$$

$$\mathbf{v}_{b,\text{ave}} = \sum_{j=1}^{\ell_b} \frac{\mathbf{v}_{b,j}}{\ell_b}, \quad \mathbf{v}_{b,\max} = \max_{j=1}^{\ell_b} \mathbf{v}_{b,j}, \quad (19)$$

$$\mathbf{v} = [\mathbf{v}_{a,\text{ave}}; \mathbf{v}_{a,\max}; \mathbf{v}_{b,\text{ave}}; \mathbf{v}_{b,\max}]. \quad (20)$$

Note that for tree composition, Equation (20) is slightly different from that in sequential composition. Our tree composition will concatenate also the hidden states computed for the roots with Equations (16) and (17), which are not shown here.

We then put $\mathbf{v}$ into a final multilayer perceptron (MLP) classifier. The MLP has a hidden layer with *tanh* activation and *softmax* output layer in our experiments. The entire model (all three components described above) is trained end-to-end. For training, we use multi-class cross-entropy loss.

**Overall inference models** Our model can be based only on the sequential networks by removing all tree components and we call it Enhanced Sequential Inference Model (**ESIM**) (see the left part of Figure 1). We will show that ESIM outperforms all previous results. We will also encode parse information with tree LSTMs in multiple layers as described (see the right side of Figure 1). We train

this model and incorporate it into ESIM by averaging the predicted probabilities to get the final label for a premise-hypothesis pair. We will show that parsing information complements very well with ESIM and further improves the performance, and we call the final model Hybrid Inference Model (**HIM**).

## 4 Experimental Setup

**Data** The Stanford Natural Language Inference (SNLI) corpus (Bowman et al., 2015) focuses on three basic relationships between a premise and a potential hypothesis: the premise entails the hypothesis (*entailment*), they contradict each other (*contradiction*), or they are not related (*neutral*). The original SNLI corpus contains also "*the other*" category, which includes the sentence pairs lacking consensus among multiple human annotators. As in the related work, we remove this category. We used the same split as in Bowman et al. (2015) and other previous work.

The parse trees used in this paper are produced by the Stanford PCFG Parser 3.5.3 (Klein and D. Manning, 2003) and they are delivered as part of the SNLI corpus. We use classification accuracy as the evaluation metric, as in related work.

**Training** We use the development set to select models for testing. To help replicate our results, we publish all our code at [xxx]. Below, we list our training details. We use the Adam method (Kingma and Ba, 2014) for optimization. The first momentum is set to be 0.9 and the second 0.999. The initial learning rate is 0.0004 and the batch size is 32. All hidden states of LSTMs, tree-LSTMs, and word embeddings have 300 dimensions.

We use dropout with a rate of 0.5, which is applied to all feed-forward connections. We use pre-trained *300-D Glove 840B* vectors (Pennington et al., 2014) to initialize our word embeddings. Out-of-vocabulary (OOV) words are initialized randomly with Gaussian samples. All vectors including word embedding are updated during training.

## 5 Results

**Overall performance** Table 1 shows the results of different models. The first row is a baseline classifier presented by Bowman et al. (2015) that considers handcrafted features such as BLEU score of the hypothesis with respect to the premise, the overlapped words, and the length difference between them, etc.

The next group of models (2)-(7) are based on sentence encoding. The model of Bowman et al. (2016) encodes the premise and hypothesis with two different LSTMs. The model in Vendrov et al. (2015) uses unsupervised 'skip-thoughts' pre-training in GRU encoders. The approach proposed by Mou et al. (2016) considers tree-based CNN to capture sentence-level semantics, while the model of Bowman et al. (2016) introduces a stack-augmented parser-interpreter neural network (SPINN) which combines parsing and interpretation within a single tree-sequence hybrid model. The work by Liu et al. (2016) uses BiLSTM to generate sentence representations, and then replaces average pooling with intra-attention. The approach proposed by Munkhdalai and Yu (2016a) presents a memory augmented neural network, neural semantic encoders (NSE), to encode sentences.

The next group of methods in the table, models (8)-(15), are inter-sentence attention-based model. The model marked with Rocktäschel et al. (2015) is LSTMs enforcing the so called word-by-word attention. The model of Wang and Jiang (2016) extends this idea to explicitly enforce word-by-word matching between the hypothesis and the premise. Long short-term memory-networks (LSTMN) with deep attention fusion (Cheng et al., 2016) link the current word to previous words stored in memory. Parikh et al. (2016) proposed a decomposable attention model without relying on any word-order information. In general, adding intra-sentence attention yields further improvement, which is not very surprising as it could help align the relevant text spans between premise and hypothesis. The model of Munkhdalai and Yu (2016b) extends the framework of Wang and Jiang (2016) to a full n-ary tree model and achieves further improvement. Sha et al. (2016) proposes a special LSTM variant which considers the attention vector of another sentence as an inner state of LSTM. Paria et al. (2016) use a neural architecture with a complete binary tree-LSTM encoders without syntactic information.

The table shows that our ESIM model achieves an accuracy of 88.0%, which has already outperformed all the previous models, including those using much more complicated network architectures (Munkhdalai and Yu, 2016b).

We ensemble our ESIM model with syntactic tree-LSTMs (Zhu et al., 2015) based on syntactic parse trees and achieve significant improvement over our best sequential encoding model ESIM, at-

| Model | #Para. | Train | Test |
|---|---|---|---|
| (1) Handcrafted features (Bowman et al., 2015) | - | 99.7 | 78.2 |
| (2) 300D LSTM encoders (Bowman et al., 2016) | 3.0M | 83.9 | 80.6 |
| (3) 1024D pretrained GRU encoders (Vendrov et al., 2015) | 15M | 98.8 | 81.4 |
| (4) 300D tree-based CNN encoders (Mou et al., 2016) | 3.5M | 83.3 | 82.1 |
| (5) 300D SPINN-PI encoders (Bowman et al., 2016) | 3.7M | 89.2 | 83.2 |
| (6) 600D BiLSTM intra-attention encoders (Liu et al., 2016) | 2.8M | 84.5 | 84.2 |
| (7) 300D NSE encoders (Munkhdalai and Yu, 2016a) | 3.0M | 86.2 | 84.6 |
| (8) 100D LSTM with attention (Rocktäschel et al., 2015) | 250K | 85.3 | 83.5 |
| (9) 300D mLSTM (Wang and Jiang, 2016) | 1.9M | 92.0 | 86.1 |
| (10) 450D LSTMN with deep attention fusion (Cheng et al., 2016) | 3.4M | 88.5 | 86.3 |
| (11) 200D decomposable attention model (Parikh et al., 2016) | 380K | 89.5 | 86.3 |
| (12) Intra-sentence attention + (11) (Parikh et al., 2016) | 580K | 90.5 | 86.8 |
| (13) 300D NTI-SLSTM-LSTM (Munkhdalai and Yu, 2016b) | 3.2M | 88.5 | 87.3 |
| (14) 300D re-read LSTM (Sha et al., 2016) | 2.0M | 90.7 | 87.5 |
| (15) 300D btree-LSTM encoders (Paria et al., 2016) | 2.0M | 88.6 | 87.6 |
| (16) 600D ESIM | 4.3M | 92.6 | 88.0 |
| (17) HIM (600D ESIM + 300D Syntactic tree-LSTM) | 7.7M | 93.5 | **88.6** |

Table 1: Accuracies of the models on SNLI. Our final model achieves the accuracy of 88.6%, the best result observed on SNLI, while our enhanced sequential encoding model attains an accuracy of 88.0%, which also outperform the previous models.

taining an accuracy of 88.6%. This shows that syntactic tree-LSTMs complement well with ESIM.

| Model | #Para. | Train | Test |
|---|---|---|---|
| (17) HIM (ESIM+syn.tree) | 7.7M | 93.5 | 88.6 |
| (18) ESIM+tree | 7.7M | 91.9 | 88.2 |
| (16) ESIM | 4.3M | 92.6 | 88.0 |
| (19) (16)-ave./max | 4.0M | 92.9 | 87.1 |
| (20) (19)-diff./prod. | 3.6M | 91.6 | 86.8 |

Table 2: Ablation performance of our best models. The difference between each pair of the results is statistically significant ($t$-test, $p < 0.01$).

**Ablation analysis** We further analyze the major components that are of importance to help us achieve good performance. From the best model, we first replace the syntactic tree-LSTM with the full tree-LSTM without encoding syntactic parse information. More specifically, two adjacent words in a sentence are merged to form a parent node, and this process continues and results in a full binary tree, where padding nodes are inserted when there are no enough leaves to form a full tree. Each tree node is implemented with a tree-LSTM block (Zhu et al., 2015) same as in model (17). Table 2 shows that with this replacement, the performance drops to 88.2%.

Furthermore, we note the importance of the layer

performing the enhancement for local inference information in Section 3.2 and the pooling layer in inference composition in Section 3.3. Table 2 suggests that the NLI task seems very sensitive to the layers. If we remove the pooling layer in inference composition and replace it with summation as in Parikh et al. (2016), the accuracy drops to 87.1%. We further remove the difference and element-wise product from the local inference enhancement layer, the accuracy drops to 86.8%.

The difference between any pairs of the models in Table 2 is statistically significant ($t$-test, $p < 0.01$). Note that in Table 1 above we do not have the output from other systems to perform a significance test, but the 0.2% difference is statistically significant among our systems. To provide some detailed comparison with Parikh et al. (2016), replacing bidirectional LSTMs in *inference composition* and also *input encoding* with MLP reduces the accuracy to 86.1% and 84.0% respectively.

**Further analysis** We showed that encoding syntactic parsing information helps recognize natural language inference—it additionally improves the strong system. Figure 3 shows an example where tree-LSTM makes a different and correct decision. In subfigure (c), the larger values at the input gates on nodes 9 and 10 indicate that those nodes are important in making the final decision. We observe

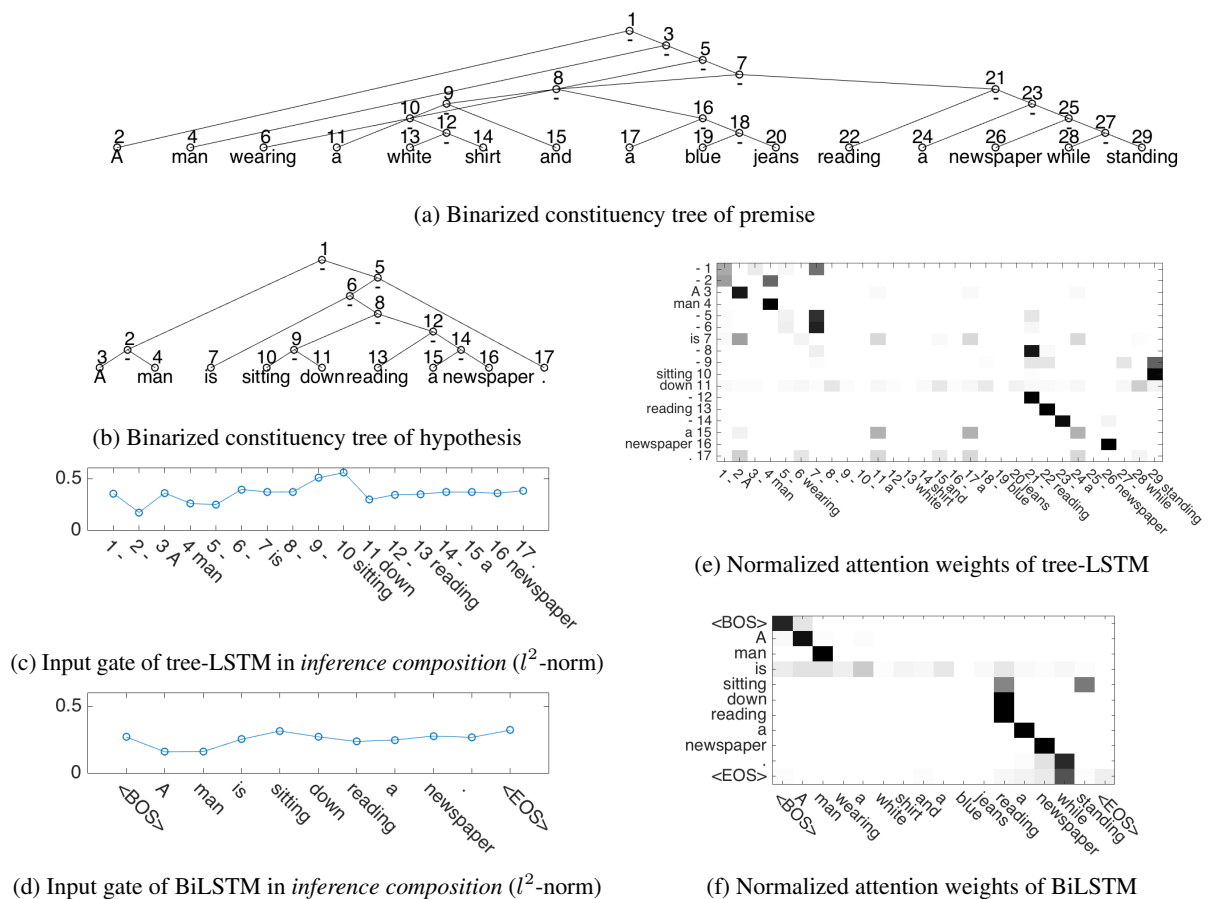

(a) Binarized constituency tree of premise

(b) Binarized constituency tree of hypothesis

(c) Input gate of tree-LSTM in *inference composition* ($l^2$-norm)

(d) Input gate of BiLSTM in *inference composition* ($l^2$-norm)

(e) Normalized attention weights of tree-LSTM

(f) Normalized attention weights of BiLSTM

Figure 3: An example for analysis. Subfigures (a) and (b) are the constituency parse trees of the premise and hypothesis, respectively. '-' means a non-leaf or a null node. Subfigures (c) and (d) are input gates' $l^2$-norm of tree-LSTM and BiLSTM in *inference composition*, respectively. Subfigures (e) and (f) are attention visualization of the tree model and ESIM, respectively. The darker the color, the greater the value. The premise is on the x-axis and the hypothesis is on y-axis.

that in subfigure (e), nodes 9 and 10 are aligned to node 29 in the premise. Such information helps the system decide that this pair is a contradiction. Accordingly, in the sequential BiLSTM, the words *sitting* and *down* do not play an important role for making the final decision. Subfigure (f) shows that *sitting* is equally aligned with *reading* and *standing* and the alignment for word *down* is not that useful.

## 6 Conclusion and Future Work

We propose neural network models for natural language inference, which achieve the best results reported on the SNLI benchmark. The results are first achieved through our enhanced sequential inference model, which outperformed the previous models, including those employing more complicated network architectures, suggesting that the potential of sequential inference models have not been fully exploited yet. Our model may serve as a new baseline or starting point for deploying more complicated architectures for NLI. Based on this, we further show that by explicitly considering recursive architectures in both local inference modeling and inference composition, we achieve additional improvement. Particularly, incorporating syntactic parsing information contributes to our best result—it further improves the performance even when added to the already very strong model.

Future work interesting to us includes exploring the usefulness of knowledge resources to help alleviate data sparseness issues. We are also interested in studying more the fragments of sentences or parses highlighted by the attention mechanism in order to provide human-readable explanations of the decisions.

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
