# Peer review of "Enhanced LSTM for Natural Language Inference"

_ACL 2017 — decision unknown_

[Official Review · Reviewer 1 · rating 4 · confidence 5]
soundness 5 · originality 5 · clarity 4 · impact 3 · substance 4 · appropriateness 5 · meaningful comparison 3 · presentation format Poster

This paper presents a purpose-built neural network architecture for textual
entailment/NLI based on a three step process of encoding, attention-based
matching, and aggregation. The model has two variants, one based on TreeRNNs
and the other based on sequential BiLSTMs. The sequential model outperforms all
published results, and an ensemble with the tree model does better still.

The paper is clear, the model is well motivated, and the results are
impressive. Everything in the paper is solidly incremental, but I nonetheless
recommend acceptance. 

Major issues that I'd like discussed in the response:
– You suggest several times that your system can serve as a new baseline for
future work on NLI. This isn't an especially helpful or meaningful claim—it
could be said of just about any model for any task. You could argue that your
model is unusually simple or elegant, but I don't think that's really a major
selling point of the model.
– Your model architecture is symmetric in some ways that seem like
overkill—you compute attention across sentences in both directions, and run a
separate inference composition (aggregation) network for each direction. This
presumably nearly doubles the run time of your model. Is this really necessary
for the very asymmetric task of NLI? Have you done ablation studies on this?**
– You present results for the full sequential model (ESIM) and the ensemble
of that model and the tree-based model (HIM). Why don't you present results for
the tree-based model on its own?**

Minor issues:
– I don't think the Barker and Jacobson quote means quite what you want it to
mean. In context, it's making a specific and not-settled point about *direct*
compositionality in formal grammar. You'd probably be better off with a more
general claim about the widely accepted principle of compositionality.
– The vector difference feature that you use (which has also appeared in
prior work) is a bit odd, since it gives the model redundant parameters. Any
model that takes vectors a, b, and (a - b) as input to some matrix
multiplication is exactly equivalent to some other model that takes in just a
and b and has a different matrix parameter. There may be learning-related
reasons why using this feature still makes sense, but it's worth commenting on.
– How do you implement the tree-structured components of your model? Are
there major issues with speed or scalability there?
– Typo: (Klein and D. Manning, 2003) 
– Figure 3: Standard tree-drawing packages like (tikz-)qtree produce much
more readable parse trees without crossing lines. I'd suggest using them.

---

Thanks for the response! I still solidly support publication. This work is not
groundbreaking, but it's novel in places, and the results are surprising enough
to bring some value to the conference.

[Official Review · Reviewer 2 · rating 3 · confidence 5]
soundness 5 · originality 5 · clarity 5 · impact 3 · substance 3 · appropriateness 5 · meaningful comparison 3 · presentation format Poster

The paper proposes a model for the Stanford Natural Language Inference (SNLI)
dataset, that builds on top of sentence encoding models and the decomposable
word level alignment model by Parikh et al. (2016). The proposed improvements
include performing decomposable attention on the output of a BiLSTM and feeding
the attention output to another BiLSTM, and augmenting this network with a
parallel tree variant.

- Strengths:

This approach outperforms several strong models previously proposed for the
task. The authors have tried a large number of experiments, and clearly report
the ones that did not work, and the hyperparameter settings of the ones that
did. This paper serves as a useful empirical study for a popular problem.

- Weaknesses:

Unfortunately, there are not many new ideas in this work that seem useful
beyond the scope the particular dataset used. While the authors claim that the
proposed network architecture is simpler than many previous models, it is worth
noting that the model complexity (in terms of the number of parameters) is
fairly high. Due to this reason, it would help to see if the empirical gains
extend to other datasets as well. In terms of ablation studies, it would help
to see 1) how well the tree-variant of the model does on its own and 2) the
effect of removing inference composition from the model.

Other minor issues:
1) The method used to enhance local inference (equations 14 and 15) seem very
similar to the heuristic matching function used by Mou et al., 2015 (Natural
Language Inference by Tree-Based Convolution and Heuristic Matching). You may
want to cite them.

2) The first sentence in section 3.2 is an unsupported claim. This either needs
a citation, or needs to be stated as a hypothesis.

While the work is not very novel, the the empirical study is rigorous for the
most part, and could be useful for researchers working on similar problems.
Given these strengths, I am changing my recommendation score to 3. I have read
the authors' responses.